# GENIE 🧜: ACHIEVING HUMAN PARITY IN CONTENT-GROUNDED DATASETS GENERATION

**Asaf Yehudai** ◇ ♣, **Boaz Carmeli** ◇, **Yosi Mass** ◇, **Ofir Arviv** ◇, **Nathaniel Mills** ◇,
**Assaf Toledo** ◇, **Eyal Shnarch** ◇, **Leshem Choshen** ◇ ♡
IBM Israel Research Lab ◇, Hebrew University of Jerusalem ♣, MIT ♡
{Asaf.Yehudai, leshem.choshen}@ibm.com

## ABSTRACT

The lack of high-quality data for content-grounded generation tasks has been identified as a major obstacle to advancing these tasks. To address this gap, we propose Genie, a novel method for automatically generating high-quality content-grounded data. It consists of three stages: (a) Content Preparation, (b) Generation: creating task-specific examples from the content (e.g., question-answer pairs or summaries). (c) Filtering mechanism aiming to ensure the quality and faithfulness of the generated data. We showcase this methodology by generating three large-scale synthetic data, making wishes, for Long-Form Question-Answering (LFQA), summarization, and information extraction. In a human evaluation, our generated data was found to be natural and of high quality. Furthermore, we compare models trained on our data with models trained on human-written data – ELI5 and ASQA for LFQA and CNN-DailyMail for Summarization. We show that our models are on par with or outperforming models trained on human-generated data and consistently outperforming them in faithfulness. Finally, we applied our method to create LFQA data within the medical domain and compared a model trained on it with models trained on other domains.

## 1 INTRODUCTION

Content-grounded generation is needed in various tasks, such as Retrieval-Augmented Generation (RAG), and content-based virtual assistants. In such tasks, the model is expected to generate a response based on a given content (i.e., information). For example, answer a question given a document that includes information needed for the answer. Zheng et al. (2023) found those types of tasks to be the second most common use cases of Language Models.

Creating datasets with elaborate responses, which rely on long content, requires an expensive and demanding manual process. This may explain why such datasets are scarce even for popular tasks such as question-answering generation. Moreover, most existing datasets were collected from noisy available resources, such as news providers (Hermann et al., 2015) and Reddit user posts (Fan et al., 2019). This lack of high-quality content-grounded data has been identified as one of the obstacles to advancing long-form QA (Stelmakh et al., 2022) and domain-specific summarization (Zhu et al., 2020), among other content-based tasks.

To address this gap, we suggest Genie, **Gen**erate **i**nformation & **e**lucidate[1], a method for creating synthetic training data for any domain and any content-grounded task. We propose a three-step process: (a) Content Preparation, (b) Generating, and (c) Filtering. Preparation is fairly straightforward, as the data may be noisy; it is best to clean it. The generation is done using a few-shot prompting approach with a large language model (LLM). See an example in App. D. Finally, since the generation is automatic, we filter its outputs to ensure their faithfulness, well-formedness, and overall quality.

Genie offers flexibility and can generate synthetic data for different domains and content-grounded generation tasks. We apply it to the tasks of long-form QA (LFQA), summarization (§ 3), and information extraction (IE) (App. C) by creating wish-QA, wish-summarization, and wish-IE. We

---

[1]We wished for a cool name and that is what we've got.

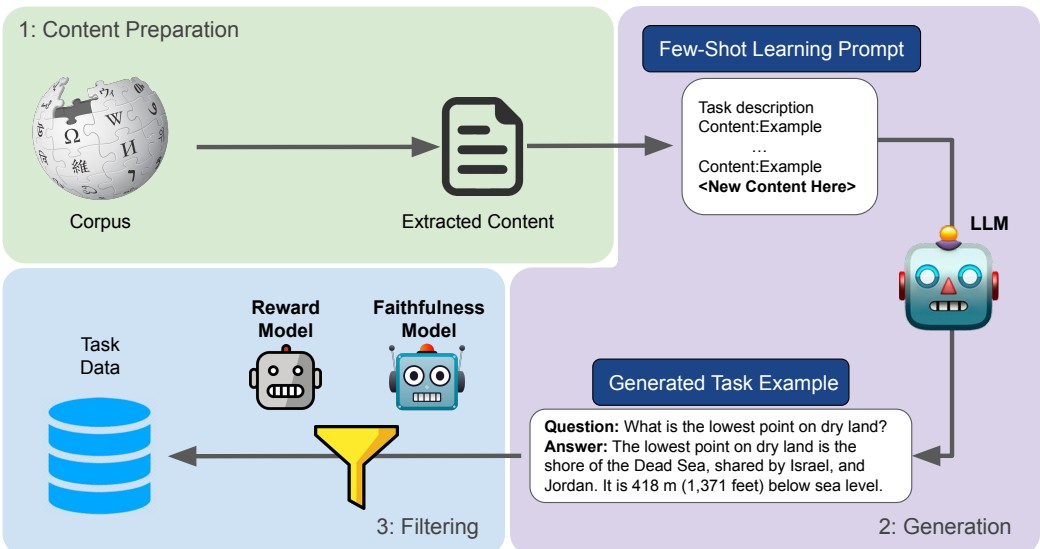

Figure 1: Genie's three steps are: (1) Extract content from the source data (2) Prompt an LLM to generate task-specific examples based on the provided content (3) filter low-quality and unfaithful examples to ensure data quality.

then show in a manual evaluation that it generates high-quality data that is natural, faithful, and lexically diverse (§4). For the task of LFQA, we compare the performance of models that were trained with wish-QA generated by Genie to those trained with the same amount of data, generated by humans (§5). We show that the former models outperform or are on par with the latter models. Additionally, faithfulness scores show that models trained on our synthetic data are more faithful to the grounding content. Those results show the overall efficacy and faithfulness of our data as training data compared to that of human-generated data. We replicate our success with summarization, showcasing the generality of the method. We publicly release all three wishes datasets.

## 2 AUTOMATICALLY CURATING DATASET FOR CONTENT-GROUNDED TASKS

In Figure 1, we illustrate Genie's three steps for automatically curating high-quality content-grounded datasets: Content Preparation, Generation, and Filtering. We refer to a content-grounded data point, like a question-answer pair or a summary, as an *example*.

### 2.1 CONTENT PREPARATION

In the preparation step, we obtain the grounding content by extracting passages from raw documents using rule-based methods. This step is the least general of our approach as it relies on the specific format in which the data is stored. If the data already exists in easy-to-use passages, it can be used as is. For example, broken by lines, found in a table, or conveniently extracted from another dataset. As the general case, we describe the extraction of content passages directly from web pages.

**Implementation details.** We crawled Wikipedia pages using browser emulation to allow dynamic content to be retrieved. Then we pass the full HTML DOM through filters to remove noise (e,g., headers, footers, sidebars, etc.). We are left with the main page content which is then transformed into Markdown, preserving the document structure (e.g., lists, tables, links, image references, articles, and sections). From this structure a table of contents is derived and based on it we break the Markdown page into passages.

## 2.2 GENERATION

In the generation step, we prompt a large language model to generate a synthetic example. For that, we use the in-context capabilities of the model. We prompt the model with four content-example pairs followed by the extracted content from the corpus with no example (see the prompt in appendix §D). The LLM generates a new example to match the extracted content.

**Implementation details.**    We decode greedily, which encourages the models to produce more grounded responses (Honovich et al., 2022b). In addition, we create two variants of the data, one by generating examples using Falcon-40B (Penedo et al., 2023) and another by generating with Llama-2-70B (Touvron et al., 2023). In general, the results using the different models have similar tendencies, with Llama being slightly better (see replications with Llama in an appendix §B). As Falcon is purely pre-trained, without additional steps we mainly report results relying on Falcon, to showcase that our method is not dependent on further alignment and instruction steps.

## 2.3 FILTERING

In the filtering step, we score each content-example pair for its format, faithfulness (i.e., grounded), and quality. For each such aspect, we implement a scoring function and filter low-scoring pairs.

**Format.**    We filter out examples where parts of the template are missing (e.g., in QA, when the prefixes signifying the start of the question or the answer are absent). Furthermore, we filter examples that are too short (less than ten words) or too long (surpassing 1.5 times the length of the grounding content for LFQA, and 0.25 for Summarization).

For QA, we use [document], [question], and [answer] as prefixes before each corresponding element. For summarization [document], [summarize], and [summary] with [summarize] representing the specific summarization instruction. It is important to note that we did not fine-tune these prompts.

**Faithfulness.**    To validate that the model-generated examples are grounded in the content, we adopt an off-the-shelf faithfulness metric and filter low-scoring examples. When deployed with trustworthy data, this can serve as a measure of correctness.

We test faithfulness by mapping the problem into a Textual Entailment (Dagan et al., 2005) or Natural Language Inference (Bowman et al., 2015) (NLI) problem. NLI involves two input sentences: a hypothesis and a premise. The objective is to determine whether the hypothesis can be inferred from the premise, contradicts it, or is neutral with respect to it. NLI models were widely utilized for faithfulness consistency evaluation (Honovich et al., 2021; Dziri et al., 2022), and most simply by taking the grounding text as the premise and the generated example as the hypothesis (Maynez et al., 2020). Here, we use the fine-tuning T5-11B NLI model presented in Honovich et al. (2022a) for assessing the generated example faithfulness.

**Quality.**    An important aspect of our methodology involves evaluating the quality of the generated examples, specifically quantifying their relevance to the corresponding task. Note that the task may be constant throughout a dataset (as is often the case of summarization) or be dependent upon an instruction (such as the question in question answering). To judge the quality automatically we use a reward model.

Reward models are trained on human preference data to give a high reward for answers that human annotators prefer. Such models can quantify quality in a human-like way, considering dimensions that are hard to isolate and measure independently by dedicated metrics. Reward models are used as quality scores for Reinforcement Learning optimization (Ouyang et al., 2022), and also serve as reference-less evaluation metrics for text generation tasks (Touvron et al., 2023).

Here, we use the reward model for both purposes and rely on the Open-Assistant model (Köpf et al., 2023), using the DeBERTa-v3 architecture (He et al., 2021). We filter generated examples whose score is below 0.5 by the reward model *reward-model-deberta-v3-large-v2*. [2] We chose 0.5

---

[2] https://huggingface.co/OpenAssistant/reward-model-deberta-v3-large-v2

as a threshold based on experimentation. Similarly, we use *t5_xxl_true_nli_mixture* [3] model to filter examples deemed unfaithful by it.

## 3  EXPERIMENTAL SETUP

Here we describe the datasets we utilized in our content-grounded generation tasks: LFQA and summarization (§3.1). Subsequently, we outline the various synthetic datasets we generated (§3.2), and finally, we discuss the models employed for training and the evaluation metrics (§3.4)

### 3.1  DATASETS

**ELI5.**  (Explain Like I'm Five) (Fan et al., 2019) comprises open-ended questions and extensive responses authored by users within the Reddit forum of the same name. To these questions and answers, retrieved documents were added as grounding content. In their manual analysis, they have found that the content is sufficient to answer $65\%$ of the questions and have information relevant to $92\%$ of the questions. In this work, we use the KILT version of the dataset (Petroni et al., 2020).

**ASQA.**  (Answer Summaries for Questions which are Ambiguous) (Stelmakh et al., 2022) is a dataset that pairs ambiguous questions from the AmbigQA dataset (Min et al., 2020) with meticulously crafted long-form answers generated through crowdsourcing. To add grounding they have used the same method presented in ELI5, but specifically retrieved documents from Wikipedia.

**NQ.**  (Natural Questions) (Kwiatkowski et al., 2019) is a dataset of real user questions sourced from the Google search engine. It includes questions and their corresponding passages (named long answers) from Wikipedia which provide potential answers and contain extractive short answers. This dataset does not have long-form answers, and here we will use only its documents for our synthetic data generation process §3.2 and will compare our synthetic questions with the questions from NQ.

**CNN-DailyMail.**  (Kwiatkowski et al., 2019) is a dataset commonly used for text summarization. It consists of news articles from CNN and the DailyMail along with their human-written summaries.

### 3.2  GENERATING SYNTHETIC DATASETS

The datasets described above were used to create datasets of synthetic data;

**Wish-QA-NQ.**  To create this dataset, we draw upon NQ passages (Kwiatkowski et al., 2019), for our synthetic data generation process. These passages are well-suited for our process because they were originally extracted from Wikipedia pages by annotators and typically consist of well-structured paragraphs, each centered around a specific topic.

**Wish-QA ELI5/ASQA.**  For the creation of a dataset that mimics the conditions of ELI5 and ASQA, where answers can be derived from multiple documents, we rely on the top three retrieved passages from either of the corresponding corpus. These passages are used as the grounding documents for constructing this synthetic dataset.

In addition, we make a new wish dataset entirely from crawled data:

**Wish-QA.**  stands for **Wi**kipedia from **S**cratc**h**[4], is a novel data we constructed following the general approach for crawling and processing as detailed in Section §2.1. It represents a realistic data generation use case from unprocessed content. We note that the extracted passages may exhibit noise and lack of coherence and conciseness.

---

[3] https://huggingface.co/google/t5_xxl_true_nli_mixture

[4] Wish-QA is also the general name for all our synthetic QA datasets

### 3.3 Models for Extrinsic Evaluation

In the Extrinsic evaluation, our goal is to compare the performance of models trained on our synthetic content-grounded data with those trained on data generated by humans. To ensure a fair comparison, we maintain an equal number of examples from each dataset (10,000) and employ identical models for training, using the same set of hyperparameters. The models we use for training are Flan-xl (Wei et al., 2021) and llama-2-13b-Chat (Touvron et al., 2023). These models serve as the foundation for facilitating comparisons across architectural variations, including Encoder-Decoder and Decoder-only models, as well as different variations of instruction fine-tuning and alignment training.

### 3.4 Evaluation Metrics

We evaluate the performance with ROUGE as a lexical similarity metric (Lin, 2004), BERT-Score as a model-based reference-based metric (Zhang et al., 2019b), and Reward model as a model-based reference-less metric. We reuse the ANLI faithfulness metric and reward mentioned in the filtering for evaluation. For faithfulness evaluation, we also calculate the K-Precision lexical similarity metric (Adlakha et al., 2023). Different performance metrics (Post, 2018; Zhang et al., 2019a, and more) showed similar results in initial trials, showing reliability of different forms (Perlitz et al., 2023).

**ROUGE.**   Following the conventional approach of assessing generated text quality, including long-form answers (Fan et al., 2019), we report the ROUGE-L score (Lin, 2004).

**BERT Score.**   (Zhang et al., 2019b) is a semantic similarity-based metric that leverages pre-trained language models to predict if the model response is semantically equivalent to the gold answer. Kasai et al. (2022) have shown BERT Score F1 is effective in evaluating many generation tasks.

**K-Precision.**   Following Adlakha et al. (2023) we report K-Precision, as it showed the highest correlation with human judgments from all lexical metrics. The metric follows the intuition, that in faithful response most words need to come from the content.

## 4 Intrinsic Evaluation

In this section, we perform intrinsic evaluation and validation of Wish-QA. We conduct a micro Turing Test, presenting synthetic and human questions side by side. We show that the questions generated synthetically are more natural than most of those found in available datasets. We also test the whole workflow and show that the filters contribute to the generated data quality and that Genie is cost and time-efficient and creates diverse data.

**Naturalness Evaluation.**   To assess the naturalness of our questions, we conducted a human-evaluation experiment. In the experiment, an expert annotator[5] was provided with two questions: one human-created and the other synthetic. Both questions were based on the same content. The annotator's task was to identify the question they believed was human-written. For this experiment, we sampled 100 questions from ELI5, ASQA, and NQ, along with their 100 synthetic counterparts.

The results in Table 1 (and App. § B) indicate that for ELI5, the synthetic question was selected as the human-written one in 72% of the cases, for NQ it was 63%, and for ASQA it was 49%. These results suggest that our synthetic questions are more natural and human-like than questions collected from sources like Reddit and Google Search engine. Additionally, they are indistinguishable from questions written by experts, such as those in the ASQA dataset. As a side finding, we also find that the ASQA dataset is of higher quality than the others, which experiments below replicate.

**Multi-Dimensional Quality Assessment.**   In this assessment, we aimed to investigate the qualities of the generated data and the impact of the filtration processes. We focused on the following dimensions: relevance and clarity of the questions, and faithfulness and overall quality of the answers. To accomplish that, we randomly selected 100 questions from the unfiltered and filtered Wish-QA. For each content-question-answer triplet, we asked annotators to answer a list of questions as shown in

---

[5]A non-author, native English speaker with an MA degree.

Table 1. The first two assessment questions aim to assess the relevance and clarity of the question. The clarity question is inspired by the findings of Min et al. (2020), which revealed that more than half of naturally occurring factoid questions are ambiguous. Following that, we include three questions related to the answer quality. These questions are designed to ascertain whether the answer adequately addresses the question while remaining faithful to the underlying content. Lastly, we ask for an overall quality rating on a 5-level Likert scale.

Human assessment results in Table 1 demonstrate that the filtration process had a significant impact on the relevance of the questions. Although our filtration setup does not directly assess the questions, we find that our faithfulness filter together with the reward filter provides an indirect signal about the relevance of the question. We also observed an improvement in the percentage of answers that were found to address the question. Faithfulness results show decent improvement, but there is still room for enhancement. Annotators' interviews reveal that despite the presence of unfaithful cases in the dataset, their granularity was often more subtle. In some instances, the model added missing pieces of information that were subsequently found to be factually correct.

We observe a slight improvement in the clarity of questions, coupled with almost all answers addressing the questions. This highlights that our answer is a single relevant response from a wide space of plausible answers, a well-documented phenomenon in LFQA (Krishna et al., 2021). Lastly, we identify an improvement in the overall score, which leads us to the conclusion that the filtering process substantially contributes to the quality and faithfulness of our dataset.

Table 1: Multi-Dimensional Quality assessment for synthetic data generated from scratch. Results show a large improvement in question relevance and the percentage of answers that address the question, answers that are faithful, and overall answer scores.

| Quality Review Question | Wish-QA w/o filters | Wish-QA w/ filters |
|---|---|---|
| Is the question **relevant** to the content? | 67% | 92% |
| Is the question **clear**? (not ambiguous) | 63% | 67% |
| Does the answer address the question? | 80% | 98% |
| Is the answer **faithful** to the content? | 53% | 76% |
| Grade the **overall quality** of the answer | 3.48 | 4.58 |

**Diversity.** Our synthetic data is built on top of large-scale content that covers many different distinct topics. As a result, our data contain diverse lexicons. We compute vocd-D (McCarthy & Jarvis, 2010) to measure the lexical diversity of our data. We found that the lexical diversity of all synthetic data is higher than their human-generated counterparts (see Table 6). We also can see that most response lengths are similar to the ones in the human writing datasets.

**Scale.** With 300K samples overall (full statistics in App. A), our dataset collection balances scale and quality. ELI5 is of a similar size but noisy, and ASQA is carefully annotated but much smaller.

**Monetary and Time Cost.** Genie is more cost-efficient and time-efficient than the traditional approach of crowd-sourced dataset curation. The cost of API's calls of models like the ones used typically ranges from $0.02 to $0.04, while the cost of an expert annotator to create a question is approximately $4.45 (Stelmakh et al., 2022). According to this rate, the 300K examples in our synthetic dataset would have cost over $1M. The time it takes to generate 10 examples is less than a minute, i.e. much faster than the time that it would take a human to read the context.

## 5 EXTRINSIC EVALUATION

Finding the synthetic data to be of high quality, we test its usefulness for improving training. We present quantitative results from our extrinsic experiments, evaluating models trained on synthetic and human-generated data on the ASQA and ELI5 test sets.

In Table 2 (and App. § B) we present Flan-xl results trained on human and synthetic data. We note that here by synthetic in-domain we refer to the case where the train and test come from the same dataset, either ELI5 or ASQA.

Results indicate that synthetic data is a competitive alternative even when human-generated data already exists. In all cases, we see substantial gains from training on the synthetic data. For example, Rouge-L almost triples from 10.5 to 28.2 for Synthetic NQ. This gain is over the already strong multitask baseline (Flan) that trained on thousands of tasks, many of which are forms of question answering.

Moreover, the synthetic data provides better or comparable results in all metrics even for cases where train and test data come from the same dataset. While – for ASQA – Rouge-L and Bert-Score are slightly lower than the in-domain training data, the synthetic data is even better than the human data on the rest of the scores on ELI5. We conclude that, if no human-generated data exists, automatically generating it has the potential to be as good.

ASQA performs better on both ASQA and ELI5 test sets. This observation implies that ASQA is, on the whole, a superior dataset compared to ELI5. This aligns with the substantial annotation efforts invested in the creation of ASQA, in contrast to the noisy and automatically scraped ELI5 data. However, it is important to note that this meticulous curation has led to a considerably smaller dataset for ASQA, totaling approximately 6k examples including the development and test sets (compared to 272k examples in ELI5). This emphasizes the contribution of our approach which allows large-scale high-quality data generation.

Another strong support for the effectiveness of our data generation approach is exemplified by the model's outputs being favored by the preference reward model, achieving comparable or higher results than the gold standard of both datasets.

Wish-QA seems to work well even with the noisy content. Wish-QA-NQ data outperformed the synthetic in-domain data across all metrics. This can be due to the quality of the Wish-QA-NQ being favorable or that a signal document generation setup is slightly preferable.

The performance on CNN-DailyMail, presented in Table 4, shows that Wish-summarization data improves upon the strong Flan-xl baseline, in Bert-Score and Reward score but not on ROUGE-L. Overall, the dataset seems comparable, attesting to the flexibility of the method.

## 5.1 Faithfulness results

Overall, Table 3 suggests training on our synthetic data leads to more content-faithful models. Models trained on Wish-QA-NQ and Synthetic data, and Wish-QA were more faithful than those trained on ASQA and ELI5 data by k-Precision and ANLI metrics. This aligns with Krishna et al. (2021), indicating LFQA models generated answers that are not grounded in the retrieved documents, and assert that this is one of the hurdles for filed progress.

Flan-xl achieves the highest Faithfulness scores followed by the synthetic datasets. Flan's achievement can be the result of its shorter and almost extractive answers. Taking into account that it is also substantially underperforming, we deduce that the synthetic datasets achieve the best trade-off across performance and faithfulness.

The faithfulness results for CNN-DailyMail are consistently high. As we observed, Flan-xl tends to produce more extractive responses. Since CNN-DailyMail primarily contains extractive summarization, it's no surprise that it exhibits high faithfulness scores. However, the model trained on our data, which doesn't emphasize extractiveness as a hard requirement, outperforms Flan-xl in terms of k-Precision, matches it in terms of NLI, and achieves the highest average level of faithfulness.

In summary, our quantitative analysis affirms that the utilization of synthetic data substantially enhances answer quality in both ASQA and ELI5 datasets. Our approach not only matches human-generated responses but also quantitatively surpasses them in terms of reward, highlighting its potential for generating higher-quality answers. Additionally, our method ensures high faithfulness and grounding in the generated responses, setting it apart from existing datasets.

## 6 Domain Adaptation

We have demonstrated that our method can generate synthetic data as good as human-generated data. Next, we hypothesize generating data directly in the target domain is more effective than from another domain for a given task. To investigate this hypothesis, we define our test set as

Table 2: Performance comparison of Flan-xl models trained on human-generated and synthetic data. The results reveal that our synthetic data consistently outperforms or achieves comparable performance to human-generated data, as indicated by ROUGE-L and Bert-Score metrics. Additionally, by reward score, models trained on our synthetic data exhibit superior or comparable performance to the gold standard responses.

| Test-set | ASQA | | | ELI5 | | |
|---|---|---|---|---|---|---|
| Train Set | ROUGE-L | Bert-Score | Reward | ROUGE-L | Bert-Score F1 | Reward |
| Flan-xl | 10.5 | 49.7 | 28.8 | 6.2 | 46.7 | 9.2 |
| ASQA | **31.4** | 66.0 | 68.6 | 13.5 | 52.2 | 24.4 |
| ELI5 | 18.7 | 58.7 | 37.2 | 13.1 | 51.3 | 11.3 |
| Wish-QA | 28.0 | **67.5** | **85.1** | **13.8** | **55.2** | 26.7 |
| Wish-QA-NQ | 28.2 | 64.8 | 80.3 | 13.2 | 54.0 | **30.3** |
| Wish-QA in-domain | 27.0 | 63.4 | 73.3 | 13.1 | 52.8 | 22.7 |
| Gold | - | - | 72.1 | - | - | **30.3** |

Table 3: Fiathfullnes performance Comparison of Flan-xl Models Trained on Human-Created and Synthetic Data. The results demonstrate that our synthetic data consistently outperforms both human-generated data and gold responses, as indicated by the k-Precision, and ANLI metrics. Flan-xl stands out with the highest scores, which can be attributed to the extractive nature of its responses.

| Test-set | ASQA | | ELI5 | |
|---|---|---|---|---|
| Train Set | k-Precision | ANLI | k-Precision | ANLI |
| Flan-xl | **98.2** | **88.7** | **89.2** | **84.9** |
| ASQA | 67.5 | 55.7 | 52.2 | 34.3 |
| ELI5 | 52.9 | 33.5 | 29.0 | 5.6 |
| Wish-QA | 77.9 | 74.9 | 58.5 | 37.9 |
| Wish-QA-NQ | 79.3 | 75.5 | 60.4 | 43.3 |
| Wish in-domain | 81.9 | 79.1 | 68.3 | 52.8 |
| Gold | 46.3 | 25.3 | 20.6 | 2.7 |

PubMed-QA, which focuses on LFQA in the medical domain. Accordingly, we create synthetic question-answering data on PubMed papers (Wish-QA-MED) as task data in the target domain. We then compare the performance of models trained on Wish-QA-MED dataset with those trained on Wish-QA-NQ data, as well as with models trained on the human-created ELI5 and ASQA datasets.

The results in Table 5 demonstrate that the synthetic dataset outperforms ELI5 and is comparable to or slightly better than ASQA in ROUGE-L and Bert-Score. Additionally, there is a more substantial gap in terms of reward and faithfulness.

Interestingly, Wish-QA-NQ and Wish-QA-MED achieve similar results, echoing the finding that Wish-QA outperforms other datasets. This suggests that out-of-domain data holds little disadvantage over in-domain data and can often surpass it. One explanation may be that providing the content with the task (e.g. QA) makes the model rely less on the training domain. Supportive evidence is the finding of Onoe et al. (2023), who found that, in their task, update strategies lag behind the performance of simply concatenating the content to the prompt. This may mean that the model relies on the content more than was previously thought (Neeman et al., 2023).

Table 4: Performance Comparison of Flan-xl Models Trained on Human-Created and Wish-Summarization Data. The results reveal that our synthetic data achieves comparable performance to human-generated data.

| Test-set | CNN-DailyMail | | | | |
|---|---|---|---|---|---|
| Train-set | ROUGE-L | Bert-Score | Reward | k-Precision | ANLI |
| Flan-xl | 30.2 | 70.9 | 96.3 | 97.6 | 98.7 |
| CNN-DailyMail | **33.3** | **72.7** | 96.5 | 97.0 | **99.1** |
| Wish-Summarization | 28.6 | 71.3 | **97.5** | **98.2** | 98.7 |

Table 5: Performance of Flan-xl Models on PubMed test data. The results reveal that our synthetic data consistently outperforms or achieves comparable performance to human-generated data in general and faithfulness metrics. Results suggest that in-domain data don't provide additional improvement for content-grounded generation, but may help the faithfulness of the model.

| Train-set | PubMed | | | | |
| --- | --- | --- | --- | --- | --- |
| | ROUGE-L | Bert-Score | Reward | K-Precision | ANLI |
| Flan-xl | 12.8 | 53.8 | 10.7 | 60.6 | 38.2 |
| ASQA | 20.5 | 61.4 | 37.3 | 77.2 | 60.8 |
| ELI5 | 15.0 | 56.3 | 16.8 | 32.2 | 2.2 |
| Wish-QA-MED | **22.1** | 61.6 | 39.4 | 78.2 | **81.8** |
| Wish-QA-NQ | 22.0 | **62.9** | **44.5** | **84.2** | 73.1 |

The faithfulness scores are inconclusive, while ANLI indicates that in-domain synthetic improves faithfulness, the k-Precision says otherwise, suggesting at least parity.

We conclude that Genie can be beneficial in creating human-level data for many tasks and domains, however, it can be that LFQA is flexible in terms of its training data domain. We leave for future research to check this finding and to show tasks or dimensions that exhibit improvement due to target domain data and can benefit from our method.

## 7 RELATED WORK

Our work is far from the first to propose synthetic data for training or experimentation (Choshen & Abend, 2019; Agarwal et al., 2020). Recently, generating data from a large language model to train a smaller one was suggested as a weak form of distillation to improve smaller models (West et al., 2022). Our method does not focus on distillation. Apart from using a stronger model for the synthetic data, those methods differ from ours as the learned model mimics a diverse set of skills, rather than becoming an expert on a task.

Still, there are a few synthetic methods for specific tasks. Most notably, methods that rely on a 2-step process, generation, and filtering. West et al. (2022) presented a 2-step pipeline for Symbolic Knowledge Distillation, rather than for creating content-grounded data. Kim et al. (2022) apply this method to create a social dialogue dataset. In Unnatural Instructions and Self-Instruct (Honovich et al., 2022b; Wang et al., 2022), they applied this method for the creation of an instruction dataset. Their method relies on model knowledge for content-grounded tasks. Similarly, Bitton et al. (2023) q2d approach uses a 2-step process for creating information-seeking dialogs. Those works share similar mechanisms with our method but differ in the content-grounded aspect of our work.

The dialog inpainting approach (Dai et al., 2022), shares a common objective with ours, to generate content-grounded question answering. They add questions between the document sentences to create a dialogue. This approach ensures the groundedness of the dialogue but it comes at the cost of less fluent and neutral conversation. In our approach, we generate the question and answer using the LLM and verify its groundedness and quality to allow both faithfulness and naturalness.

## 8 DISCUSSION

Our work introduces Genie, an efficient and cost-effective automated approach for curating content-grounded datasets. Our method incorporates a novel filtering mechanism to ensure data quality. We demonstrate that our synthetic wish-QA and wish-summarization data achieves parity with expert human datasets in both intrinsic and extrinsic evaluations. Furthermore, we illustrate that our data surpasses human-written datasets in terms of lexical diversity and faithfulness. We have also proven the applicability of our method to noisy crawled data.

We want to emphasize the immense potential this approach holds for facilitating the development of content-focused datasets and, consequently, generative models, minimizing the need for costly human annotation. Therefore, our method democratizes the creation of such datasets and models, making them more accessible to the entire community.

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

## A  DATASETS STATISTICS

While the amount of synthetic data is capped merely by the actual amount produced and the content available, meaning the others can generate more data, we supply the actual sizes we release. Statistics are shown in Table 6

## B  COMPLEMENTARY RESULTS.

We provide additional human experiments (Table 7) demonstrating that using clean content from exciting datasets can further improve the synthetic data quality.

We also provide the results of synthetic datasets created with the stronger Lamma model in Table 8 and on PubMed on Table 9.

We also include here results with Llama-2-13B-Chat as the base model. General results are presented in Table 11 and Faithfulness results are in Table 10.

Table 6: Data statistic for several datasets including the size of the data, the number of words in the response, and the average lexical diversity. Overall, our dataset spans 300K+ samples, similar to the size of ELI5, and CNN-DailyMail which were collected from available resources and are noisy by nature. On the other hand, our data is 50 times larger than the carefully annotated ASQA data. We also can see that most response lengths are similar to the ones in the human writing datasets, and the lexical diversity of all synthetic data is higher than their human writing counterparts.

|  | # samples | # words in response | lexical diversity |
|---|---|---|---|
| ELI5 | 272K | 107.0 | 70.9 |
| ASQA | 6,316 | 73.3 | 65.8 |
| CNN-DailyMail | 311K | 44.8 | 40.1 |
| Wish | 27K | 65.1 | 54.3 |
| Wish-QA-NQ | 88K | 73.4 | 62.3 |
| Wish-QA-ELI5 | 78K | 98.3 | 71.8 |
| Wish-QA-ASQA | 6140 | 87.5 | 67.0 |
| Wish-Summarization | 37K | 66.6 | 76.0 |
| Wish-QA-Med | 69K | 84.2 | 71.8 |

Table 7: Human assessment for Wish-QA without and with filtering, and Synthetic NQ. Results show a big improvement in question relevance and the percentage of answers that address the question, answers that are faithful, and overall answer scores.

| Quality Review Question | Wish-QA w/o filters | Wish-QA w/ filters | Synthetic NQ |
|---|---|---|---|
| Is the question **relevant** to the content? | 67% | 92% | 98% |
| Is the question **clear**? (not ambiguous) | 63% | 67% | 91% |
| Does the answer address the question? | 80% | 98% | 98% |
| Is the answer **faithful** to the content? | 53% | 76% | 88% |
| Grad the **overall quality** of the answer | 3.48 | 4.58 | 4.74 |

## C    INFORMATION EXTRACTION RESULTS.

To test the efficacy of the method in another task, we conducted an experiment focusing on information extraction (IE). For this experiment, we utilized the information extraction part of the Databricks dataset , which we divided into train and test sets (1000/50). Synthetic information extraction data was generated over Google-NQ passages, requiring only a new set of few-shot examples to demonstrate the task, without any other modifications. The table below 12 shows that our model improved in all metrics compared to the Flan-xl baseline, except for ANLI. Furthermore, it achieves higher scores than the model trained on human data in Reward and K-Precision, while slightly lagging behind in ROUGE-L, Bert-Score, and ANLI. The average of the faithfulness metrics indicates comparable performances in this dimension for the models trained on human and synthetic data. These experiments reinforce our belief that our method is general and can be readily applied to various tasks, yielding improved results through the generation of high-quality synthetic data.

Table 8: Results on ASQA and ELI5 including models trained with Llama-2-70B-chat synthetic data.

| Test-set | ASQA | | | ELI5 | | |
|---|---|---|---|---|---|---|
| Train Set | ROUGE-L | Bert-Score F1 | Reward | ROUGE-L | Bert-Score F1 | Reward |
| Flan-xl | 10.5 | 49.7 | 28.8 | 6.2 | 46.7 | 9.2 |
| Human in-domain | **31.4** | 66.0 | 68.6 | 13.1 | 51.3 | 11.3 |
| Human out of domain | 18.7 | 58.7 | 37.2 | 13.5 | 52.2 | 24.4 |
| Wish-QA | 28.0 | **67.5** | **85.1** | **13.8** | **55.2** | 26.7 |
| Wish-QA-NQ Falcon | 28.2 | 64.8 | 80.3 | 13.2 | 54.0 | 30.3 |
| Wish-QA-NQ Llama | 28.1 | 64.8 | 81.7 | 13.6 | 53.9 | **44.1** |
| Wish-QA in-domain Falcon | 27.0 | 63.4 | 73.3 | 13.1 | 52.8 | 22.7 |
| Wish-QA in-domain Llama | 28.0 | 64.6 | 76.4 | 13.7 | 53.6 | 30.5 |
| Gold | - | - | 72.1 | - | - | 30.3 |

Table 9: Full results on PubMed dataset. Here we include results where Llama-2-70B is the data generator.

| Train-set | PubMed | | | | |
| | ROUGE-L | Bert-Score | Reward | k-Precision | ANLI |
|---|---|---|---|---|---|
| Flan-xl | 12.8 | 53.8 | 10.7 | 60.6 | 38.2 |
| ASQA | 20.5 | 61.4 | 37.3 | 77.2 | 60.8 |
| ELI5 | 15.0 | 56.3 | 16.8 | 32.2 | 2.2 |
| Wish-QA-Med Falcon | **22.1** | 61.6 | 39.4 | 78.2 | 81.8 |
| Wish-QA-NQ Falcon | 22.0 | **62.9** | 44.5 | **84.2** | 73.1 |
| Wish-QA-Med Llama | 21.4 | 62.1 | 30.0 | 74.6 | **92.4** |
| Wish-QA-NQ Llama | 21.4 | 62.5 | **51.2** | 80.6 | 83.0 |

Table 10: Faithfulness results on ASQA and ELI5 when Llama-2-13b-Chat is the base model.

| Test-set | ASQA | | ELI5 | |
| Train-set | k-Precision | ANLI | k-Precision | ANLI |
|---|---|---|---|---|
| Llama-13b-chat | 40.4 | 65.5 | 23.3 | 24.6 |
| ASQA | 77.3 | 73.1 | 48.0 | 37.3 |
| ELI5 | 59.0 | 42.6 | 33.3 | 13.8 |
| Wish-QA-NQ | **81.1** | **76.2** | 58.2 | 45.1 |
| Wish-QA-ASQA | 79.2 | 73.1 | 51.8 | 40.3 |
| Wish-QA-ELI5 | 77.6 | 75.9 | **52.0** | **46.1** |

Table 11: Results on ASQA and ELI5 when Llama-2-13b-Chat is the base model.

| Test-set | ASQA | | | ELI5 | | |
| Train-set | RougeL | Bert-Score | Reward | RougeL | Bert-Score | Reward |
|---|---|---|---|---|---|---|
| Llama-13b-chat | 21.4 | 57.6 | 62.7 | 11.8 | 51.2 | 46 |
| ASQA | **29.0** | **65.2** | **91.1** | 12.4 | **53.5** | **66.9** |
| ELI5 | 13.4 | 55.0 | 86.4 | 11.4 | 52.8 | 66.7 |
| Synthetic NQ | 26.3 | 64.4 | 86.9 | **12.5** | **53.5** | 52.6 |
| Synthetic ASQA | 26.3 | 63.6 | 87.1 | 12.3 | 53.1 | 57.7 |
| Synthetic ELI5 | 26.3 | 63.6 | 89.2 | 12.4 | 53.0 | 57.1 |

Table 12: Information Extraction results on the Databricks dataset.

| Model | ROUGE-L | Bert-Score | Reward | K-Precision | ANLI |
|---|---|---|---|---|---|
| Flan-xl | 42.3 | 67.3 | 35.9 | 39.1 | 89.1 |
| Databricks | 54.6 | 77.2 | 70.1 | 56.7 | 89.9 |
| Wish-IE | 47.9 | 74.5 | 87.4 | 60.0 | 86.8 |

# D    PROMPT

We provide examples of the prompts used to generate data in Fig.2.

Figure 2: An illustration of our data generation prompt. In black is the few-shot prompt we give the model. In pink a new QA that the model generated based on the the provided content.

**Instruction:** Given the next [document], create a [question] and [answer] pair that are grounded in the main point of the document, don't add any additional information that is not in the document. The [question] is by an information-seeking user and the [answer] is provided by a helping AI Agent.

**[document]:** Scrumptious Sweet Co. factory ..

**### Response:**
**[question]:** Where was the movie Chitty Chitty Bang Bang filmed?
**[answer]:** Chitty Chitty Bang Bang was filmed in a variety of locations in England and Germany. Some ..

**[document]:** The series stars Eugene Levy as Johnny Rose …

**### Response:**
**[question]:** What is the plot of the show Schitt's Creek?
**[answer]:** The show Schitt's Creek is about a wealthy family who loses their fortune and is forced to rebuild their lives in a small town. The show follows the family as they adjust to their new life in the town and learn to appreciate the simple things in life.

**[document]:** 2016's countdown broke several Hottest 100 records…

**### Response:**
**[question]:** What was the most popular song on the 2016 Hottest 100?
**[answer]:** The most popular song on the 2016 Hottest 100 was "Never Be Like You" by Flume. This was the first time that an electronic dance music producer topped the countdown.

**[document]:** In Greek mythology, Persephone ..

**### Response:**
**[question]:** Who is Persephone in Greek mythology and what is her significance?
**[answer]:** In Greek mythology, Persephone, also known as Kore, is the daughter of Zeus and Demeter. She ..

**[document]:** Terry, the dog that played Toto in the 1939 screen …

**### Response:**
**[question]:** What breed of dog was Toto in the Wizard of Oz?
**[answer]:** Terry, the dog that played Toto in the 1939 screen adaptation of The Wizard of Oz, was a brindle Cairn terrier.

