# OpenReview forum: "Achieving Human Parity in Content-Grounded Datasets Generation"
_ICLR.cc/2024/Conference — ICLR 2024 poster_

### Official Review · Reviewer_Y1CY · 2023-10-28

**Soundness:** 2 fair
**Presentation:** 2 fair
**Contribution:** 2 fair
**Rating:** 5
**Confidence:** 4

**Summary:**

This paper proposes a method for automatically generating high-quality content-grounded datasets for tasks such as question answering and summarization. The method consists of three stages: content preparation, generation, and filtering. The authors showcase the effectiveness of their methodology by generating large-scale data for synthetic long-form question answering and summarization tasks. They compare models trained on their synthetic data with models trained on human-generated data and show that their models perform equally or better in terms of quality and faithfulness.

**Strengths:**

1. The paper presents an innovative and practical methodology for generating content-grounded datasets. The three-stage process of content preparation, generation, and filtering provides a systematic approach to ensure the quality and faithfulness of the generated data.
2. The authors provide insightful empirical findings by comparing models trained on their synthetic data with models trained on human-generated data.

**Weaknesses:**

1. Content-grounding generation is a broad topic, and the paper only mentions long-form QA and summarization. However, if we look at it purely from the perspective of "Content-grounding generation," most traditional NLP tasks can be transformed into the paradigm: Instruction+Input -> Output, which matches with the definition of Content-grounding generation. However,  the paper only validates this on two tasks. Furthermore, in subsequent experiments, the paper only constructs data based on specific datasets, raising doubts about the generality of the proposed method. Therefore, I tend to believe that the paper exaggerates its contributions.
2. The data construction methods mentioned in the paper rely on the format and source (Wiki) of previous datasets, making it challenging to assess their practicality in an open-world context.

**Questions:**

Please check the weakness.

---

> ### Author Response · Authors · 2023-11-16
> **Addressing Concerns and Highlighting Robustness in Content-Grounded Generation Methodology**
>
> Thank you very much for your helpful review. We are glad you find our method innovative and practical, and our filtering to be a systematic approach ensuring the quality and faithfulness of the data. We also appreciate you find the experimental comparison of models trained on our synthetic data with models trained on human-generated data to be insightful and empirically sound.
>
> 1. We completely agree that Content-grounding generation is a broad topic. We chose two permanent tasks, LFQA, and summarization, as their representative. For instance, the Instruction data for the Dolly model consisted of three content-grounded tasks, two of which are LFQA and summarization (https://www.databricks.com/blog/2023/04/12/dolly-first-open-commercially-viable-instruction-tuned-llm ). The third task, information extraction, is highly related to LFQA and can be viewed as an extractive version of LFQA. We demonstrated that our method is robust to different setups for the LFQA task, including multi-document and single-document, data collected from scratch and data from existing datasets, data from Wikipedia and data from the medical domain, and additional practical cases (which we could not report due to data privacy). Moreover, we showed that our method is also robust for different generation models and trained models. Consequently, we see no reason why our method would not hold for other content-grounded tasks like information extraction or paraphrasing. Additionally, to clear this concern, we plan to demonstrate that our method remains effective for additional tasks and other datasets to establish its generality beyond the initial two tasks.
>
> 2. Regarding the claim that our data construction method is limited, as we explained in the paper, we have a general approach for content preparation that uses a set of pages and prepares them at a passage level, ready for the two subsequent steps in the data generation process. We demonstrated this setup in the creation of the WiFS dataset (Wikipedia From Scratch). This step is general, and we applied it to several domains, including banking, mobile telecom, sports, and IT support (practical cases where the data is private). Regarding the use of previous datasets, we used them mostly to facilitate fair comparison. We are not limited to Wikipedia, as we also used PubMed and CNN-DailyMail. In most cases, those datasets were processed automatically. Google-NQ is the only dataset where human annotators were involved in extracting the passage from Wikipedia, but our setup did not require the use of questions, making the contribution of human annotators less crucial. Furthermore, the model trained on the synthetic WiFS data performed equally or better in terms of quality and faithfulness than the one trained on Google-NQ, indicating that the Google-NQ content preparation setup is not crucial for the success of our method. We will make this point clearer in the final version.

---

> ### Author Response · Authors · 2023-11-20
> **Information Extraction Task experiments, More Wifs Results**
>
> To address concerns about the limitation of our method to question answering and summarization, we conducted a new experiment focusing on information extraction (IE). For this experiment, we utilized the information extraction part of the Databricks dataset, which we divided into train and test sets. Synthetic information extraction data was generated over Google-NQ passages, requiring only a new set of few-shot examples to demonstrate the task, without any other modifications. The table below shows that our model improved in all metrics compared to the Flan-xl baseline, except for ANLI. Furthermore, it achieves higher scores than the model trained on human data in Reward and K-Precision, while slightly lagging behind in ROUGE-L, Bert-Score, and ANLI. The average of the faithfulness metrics indicates comparable performances in this dimension for the models trained on human and synthetic data. These experiments reinforce our belief that our method is general and can be readily applied to various tasks, yielding improved results through the generation of high-quality synthetic data.
>
> | IE          | ROUGE-L | Bert-Score | Reward | K-Precision | ANLI  |
> |-------------|---------|------------|--------|-------------|-------|
> | Flan-xl     | 42.3    | 67.3       | 35.900 | 39.1        | 89.1  |
> | Databricks  | 54.6    | 77.2       | 70.100 | 56.7        | 89.9  |
> | Synthetic NQ - IE  | 47.48   | 74.5       | 87.400 | 60.0          | 86.8  |
>
>
>
> Additionally, addressing concerns about experiments conducted solely with data from specific datasets, we present the experiment results of Wifs on the PubMed test set. The results indicate that Wifs scores the highest in ROUGE-L and Bert-Score, performs well in K-Precision and ANLI, and outperforms Flan-xl and ELI5 in the Reward score. However, it achieves lower scores than ASQA and other synthetic datasets.
>
> | PubMed Train-set    | ROUGE-L | Bert-Score | Reward | K-Precision | ANLI  |
> |---------------------|---------|------------|--------|-------------|-------|
> | Flan-xl              | 12.8    | 53.8       | 10.7   | 60.6        | 38.2  |
> | ASQA                | 20.5    | 61.4       | 37.3   | 77.2        | 60.8  |
> | ELI5                | 15.0    | 56.3       | 16.8   | 32.2        | 2.2   |
> | Synthetic PubMed    | 22.1    | 61.6       | 39.4   | 78.2        | 81.8  |
> | Synthetic NQ        | 22.0    | 62.9       | 44.5   | 84.2        | 73.1  |
> | Wifs                | 23.2    | 69.0       | 29.7   | 81.7        | 67.3  |
>
> We believe and hope this addresses both of the proposed weaknesses showing those were merely a question of the amount of experimentation rather than a flaw in the method, which is now addressed.

---

### Official Review · Reviewer_pq6E · 2023-10-29

**Soundness:** 2 fair
**Presentation:** 2 fair
**Contribution:** 2 fair
**Rating:** 5
**Confidence:** 3

**Summary:**

The paper introduces a method for the automated generation of high-quality, content-grounded data through a three-stage process. The experiments reveal that models trained on this data either match the performance or surpass models trained on human-generated data, with an advantage in terms of faithfulness.

**Strengths:**

1. Employing Large Language Models (LLMs) to generate high-quality datasets is both logical and intuitive.
2. Experiments showcase the method's effectiveness for both QA and summarization tasks.

**Weaknesses:**

1. The paper claims that high noise levels in existing datasets, such as news domains. This claim necessitates empirical validation. Additionally, the test set originates from the same noisy source. It is necessary to perform a Multi-Dimensional Quality assessment on both the original and synthetic datasets to evaluate genuine quality enhancements. The paper lacks specifics on annotation details, including inter-annotator agreements.
2. What if using llama 70b chat synthetic data to train a llama 70b model?  If we're talking about cost and time efficiency, then using synthetic data from LLMs to train smaller models might not be the most economical approach, if LLM itself can work well. The paper argues about the method being more cost-effective than traditional crowd-sourced dataset curation, which might exceed $1M. Yet, directly sourcing text from the web can be free, and existing filtering methods can be used to ensure data quality.
3. The paper is more like an empirical exploration into using LLMs for synthetic data generation, which is not inspiring for other works.

**Questions:**

why using synthetic data can improve performance on noisy test sets?

---

> ### Author Response · Authors · 2023-11-16
> **Addressing the claim of noisy test sets, adding annotation details, outperforming Llama-2-13b-Chat, and the need for our method over using data from the internet.**
>
> Thank you for your valuable review. We are glad you find our paper well-written, our method effective in generating human-level data, our evaluations to be thorough, and our work to be technically sound.
>
> 1. Regarding providing support for the claim that existing datasets have high noise levels. We state that the process used to collect a large amount of content-grounded data is noisy as it relies on some assumption, e.g. the answer/summary is based on the documents, and the answer adequately addresses the question. This was found to be a main problem for both LFQA and summarization. For LFQA, https://aclanthology.org/2021.naacl-main.393/ found that large numbers of  ELI5 answers are not faithful to the documents and described it as one hurdle for advancing the task of LFQA. In summarization, https://aclanthology.org/2023.findings-acl.220/ describes that “models still suffer from generating factually inconsistent summaries, reducing their utility for real-world application” in regard to the XLSum news summarization dataset. We can add that this is not always the case as the CNN-DailyMail dataset seems to be very faithful by reviewing the faithfulness scores of models trained on it. Accordingly, We base our claim that datasets created with this process can be noisy on the conclusions of the cited papers. We will make sure it is clearer in the paper's final version.
>
> Regarding the test being noisy, you are right that there is a problem with the assessment on those test sets. First, we note that for the LFQA test set, this problem is relevant for the ELI5 test, while the ASQA was human writing and the PubMed was humanly validated alleviating the concern of them being noisy. Therefore, when seeing that the different test sets share similar trends (for example, the best-performing model is mostly the same; Tables 2 & 3) we get an additional indication of the validity of our results. For the summarization test set, the CNN-DailyMail seems mostly faithful making this concern less severe. We will add this clarification to the paper.
>
> Regarding annotation details, two expert annotators participated in the study. In the Naturalness Evaluation experiments, the maximum discrepancy between their assessments was 6%. The inter-annotator agreements for ELI5, Google-nq, and ASQA evaluations are 70% (Cohen's Kappa: 0.40), 64% (Kappa: 0.28), and 57% (Kappa: 0.14), respectively. These figures are indicative of the varying degrees of difficulty in distinguishing between human and model writing across different datasets. In the Multi-Dimensional Quality assessment, the greatest difference observed per question annotation was 7%, and the average overall quality difference was 0.17. There was a slightly higher variation for WiFS w/o filters and a lower difference for Synthetic NQ. We will also add this to the paper.
>
> 2. We are uncertain about our complete comprehension of this point. To the best of our understanding, the question is: given that synthetic data was generated with llama-70b-chat, why not utilize it to train a llama-70b (non-chat model)? If the intention is to distill the model to a smaller size, there are alternative methods that may be more effective than distillation using synthetic data. Firstly, it's important to note that this question is specifically relevant to the data produced with llama-70b-chat and is not applicable to Falcon-40b, which did not undergo instruction fine-tuning. Secondly, we refer you to tables 10 and 11 in Appendix B, where we demonstrate that our method surpasses Llama-2-13b-Chat in general metrics and faithfulness by significant margins. This highlights the critical role of our filtering method in enhancing our data quality and faithfulness.
> Concerning the assertion that "sourcing text from the web can be free," in general, this holds true. However, when dealing with content-grounded data, it is constrained by the availability of existing data, unless, of course, you adapt the available data to become content-grounded, which is exactly what we propose. For most domains and tasks, finding such datasets on the internet is challenging. Our method empowers the creation of datasets for any domain and content-grounded task, as exemplified in LFQA in the medical domain.
> As for the mention of "existing filtering methods," which method are you referring to? To our knowledge, most previous works did not employ post-filtering methods for quality and faithfulness (see ELI5 and XLSum). If you are alluding to our filtering approach, it is indeed an integral part of our method and contribution. We acknowledge its applicability in automatically enhancing existing datasets. Furthermore, relying solely on the filtering mechanism ties you to the available domain and task of the data on the internet. Our approach provides the flexibility to curate data for any content-grounded task and domain with available data.

---

> ### Author Response · Authors · 2023-11-16
> **The Promise of Synthetic Content-Grounded Data Generation**
>
> 3. You are surely entitled to this opinion (is such work against the guidelines?), however, we do not share it. Synthetic data generation is an important field of research that showed promising results in improving model abilities. From that perspective, our work shows a novel method to create high-quality, content-grounded data which is cost and time-efficient improves over strong basslines, and shows that our data achieves human level which was not shown before. Other than that, the data itself has been a bottleneck to developing content-grounded models, and our opinion is hence that the released data and method would enable such work in other domains that lack this data.

---

> > ### Author Response · Authors · 2023-11-20
> > **Multidimensional Comparison**
> >
> > Regarding your suggestion for a multidimensional evaluation of ELI5 and ASQA, including both human and synthetic data, we conducted an experiment using the same setup as described in the paper. The results are presented in the table below, and we will incorporate them into the paper:
> >
> > | Quality Review Question                           | ELI5  | ASQA  | ELI5 - synthetic | ASQA - synthetic |
> > |---------------------------------------------------|-------|-------|------------|------------|
> > | Is the question relevant to the content?           | 77%   | 100%  | 100%       | 99%        |
> > | Is the question open-ended? (not ambiguous)             | 31%   | 23%   | 57%        | 54%        |
> > | Does the answer address the question?              | 92%   | 96%   | 100%       | 98%        |
> > | Is the answer faithful to the content?             | 15%   | 72%   | 95%        | 92%        |
> > | Grad the quality of the answer (1-5)               | 3.98  | 4.88  | 4.74       | 4.73       |
> >
> > In the table, it is evident that, except for ELI5, the majority of questions were relevant to the content. Many questions, particularly in ELI5, are non-factoid with multiple possible answers. In all datasets, answers generally address the questions. Regarding faithfulness to the content, ELI5 lags behind with only 15% of cases being faithful, while ASQA performs better but still not as faithfully as our synthetic data. Overall, the answer quality of ASQA appears to be the highest, slightly surpassing the scores of the synthetic data. Both significantly outperform the scores of ELI5.

---

### Official Review · Reviewer_xd6k · 2023-10-31

**Soundness:** 3 good
**Presentation:** 3 good
**Contribution:** 2 fair
**Rating:** 6
**Confidence:** 3

**Summary:**

The paper focuses on LLM-based synthetic data generation for content-based generation tasks (such as long-form QA).
The paper uses LLMs with in-context task-specific examples and new contents to generate synthetic content-related data (eg. question-answer pairs). This data is then filtered based on various factors (reward model scores, format etc.). Through multiple evaluations certain synthetic data created in that form is found to be on par with human data.

**Strengths:**

1. The paper is well written.
2. The paper demonstrates that the synthetic data generated through LLM can achieve parity with human data) for content-grounded generation tasks.
3. Multiple forms of evaluations are done.

**Weaknesses:**

1. While I have not noticed any technical issues with the paper, the main weakness seems to be the novelty and the scope of the contribution. The 2-step based generation seems fairly obvious as an approach, and it seems already been tried before in a few works as elaborated in the related works. While the exploration of content-grounded generation tasks may be new, I am not sure if simply using prior methods in another task setup (without tackling any fundamental task-specific challenges not relevant in prior works) is enough to meet the bar for ICLR.

**Questions:**

I am open to increasing the score if I am missing some context, highlights, or critical contrast compared to prior works. This could be provided in the rebuttal.

---

> ### Author Response · Authors · 2023-11-16
> **Innovative Solutions for Content-Grounded Data Generation**
>
> Thank you very much for your helpful review. We are glad you find our paper well-written, our method effective in producing human-level data, our evaluations to be thorough, and our paper to be technically sound.
>
> Regarding the scope of our contribution, we agree that the 2-step-based generation is indeed an intuitive, effective, and broad approach with many different realizations for different tasks. Our work utilizes this approach in an innovative way,  as highlighted by R3,  to address a real-world challenge identified in previous works as well as practical applications.
>
> Our method tackles the real-world problem of limited faithful content-grounded data which hinders the usability of trained models in real-world applications. In response, we present an efficient method to generate such data, providing a general approach for creating such data from any given content.  Our results demonstrate that models trained on our data not only match but often surpass the performance of models trained on human data, with a larger advantage in terms of faithfulness.
>
> High-quality content-grounded data is scarce (https://arxiv.org/abs/2204.06092) as large datasets are limited in their faithfulness (https://aclanthology.org/2021.naacl-main.393/). This scarcity calls for ways to create such datasets.
> Previous works for content-grounded data generation either created a large dataset with limited naturalness (dialogue inpainting) or paid a lot to annotators to create a fairly small (6k) dataset (e.g., ASQA).
>
> Additionally, most previous methods are not fitting to generate knowledge-based faithful data. Most methods rely on the LLMs' parametric knowledge as a source of information without verifying its validity.
>
>  In contrast, our method enables the creation of natural and faithful task-specific data grounded in specific content, offering a solution for tasks and domains with limited data availability. Moreover, it paves the way for comprehensive pipelines that span from scraped data to task-specific datasets, showcasing significant potential.
>
> It's worth noting that our method was specifically designed to address a problem we encountered, emphasizing its practical relevance and targeted efficacy. We appreciate your feedback and will make sure that our significant contribution to advancing the field of content-grounded data generation will be clearer.

---

> ### Comment · Reviewer_xd6k · 2023-11-17
>
> Thank you for the feedback. All in all, I have decided to increase my score to 6.
>
> I still think the novelty and the "insight" factors are a bit borderline given that "similar enough" (by my standards) techniques have been already used and introduced in prior works for synthetic data generation. I will leave that to the meta-reviewer and SAC as to how to factor that in.
>
> Nevertheless, I can see the practical value of the paper for the community of LFQA - which as you said has a scarcity of good large-scale datasets. Moreover, there could be also potential value in using cheap somewhat noisy synthetic data for pre-training or some "intermediate stage" training before fine-tuning in more high-quality data.
>
> While I think enough evidence has been provided to demonstrate the practical value of the dataset generation method for LFQA, the paper can be still improved from the suggestions of Reviewer pq6E with comparable multidimensional evaluations on some of the baseline datasets (ELI5, ASQA etc.). It would be also good if you could present a table here with the inter-annotator agreements of all relevant datasets side-by-side.
>
> I also realized that the related works can be potentially improved with other works on creating query-focused summarization, and non-factoid QA. I will cite three recent works here [1,2,3], but it can be extended with a deeper literature review from the citation network.
>
> [1] ANTIQUE: A Non-factoid Question Answering Benchmark - Hashemi et al. ECIR 2020
>
> [2] WikiHowQA: A Comprehensive Benchmark for Multi-Document Non-Factoid Question Answering - Bolotova et al. ACL 2023
>
> [3] LMGQS: A Large-scale Dataset for Query-focused Summarization - Xu et al. ArXiv 2023
>
> Also, will the full datasets be released?

---

> ### Author Response · Authors · 2023-11-20
> **Multidimensional Comparison, Data Release**
>
> Thank you very much for acknowledging our contribution to the LFQA community and deciding to increase the score.
> We are still a bit unsure about what you consider similar techniques. As far as we know, no prior works on synthetic data generation have used either reward models or models for faithfulness detection. These two approaches have been shown to be effective in improving the quality of our synthetic content-grounded data. Moreover, most previous works heavily rely on instruct-tuned proprietary models, limiting their usability.
> Regarding the suggestion of reviewer pq6E for a multidimensional comparison of ELI5, ASQA, human, and synthetic data, we conducted this experiment with the same setup as described in the paper. Here are the results, which we will add to the paper:
>
>
> | Quality Review Question                           | ELI5  | ASQA  | ELI5 - synthetic | ASQA - synthetic |
> |---------------------------------------------------|-------|-------|------------|------------|
> | Is the question relevant to the content?           | 77%   | 100%  | 100%       | 99%        |
> | Is the question open-ended? (not ambiguous)             | 31%   | 23%   | 57%        | 54%        |
> | Does the answer address the question?              | 92%   | 96%   | 100%       | 98%        |
> | Is the answer faithful to the content?             | 15%   | 72%   | 95%        | 92%        |
> | Grad the quality of the answer (1-5)               | 3.98  | 4.88  | 4.74       | 4.73       |
>
> In the table, you can see that except for ELI5, the majority of questions were relevant to the content. Indeed, many questions are non-factoid questions with multiple possible answers. In all datasets, the answers address the questions in most cases. Regarding the faithfulness of the answers to the content, ELI5 is far behind with only 15% of cases being faithful to the content, while ASQA is much better in that aspect but still not as faithful as our synthetic data. These results are in line with our expectations, such as the faithfulness automatic metrics results of the fine-tuned models and the ASQA paper manual analysis (if taking into account our use of top-3 passages), which suggest a similar discrepancy in dataset quality. Finally, the overall answer quality of ASQA appears to be the best, slightly higher than the synthetic data scores. Both significantly outperform the ELI5 scores.
> Regarding the inter-annotator agreement scores, here is a table with their scores:
>
> | Naturalness Evaluation Experiment | Maximum Discrepancy | Inter-Annotator Agreement (%) | Cohen's Kappa |
> |------------------------------------------|---------------------|-------------------------------|---------------|
> | ELI5             | 5%                  | 70%                           | 0.40          |
> | Google-nq        | 6%                  | 64%                           | 0.28          |
> | ASQA             | 6%                  | 57%                           | 0.14          |
>
> Regarding the additional relevant papers about query-focused summarization and non-factoid QA, they indeed seem relevant, and we will add them. We will also look for more works on those topics that are relevant.
> Regarding the release of the dataset, as we mentioned in the paper (just before Section 2), we plan to publicly release the data.

---

### Meta-Review · Area_Chair_x4Tu · 2023-12-05

**Metareview:**

The reviewers appreciate the usefulness of the approach but also point out the limited novelty, noting that the idea has been tried in prior work, the fact that this method may not be as cost-effective as the authors claim, and the generalizability to broader tasks.

However, I felt that the authors did a good job in answering the reviewers' questions and also provided additional requested experiment results. I would therefore recommend accepting this as a poster.

**Justification For Why Not Higher Score:**

Given that only one reviewer gave a 6, I am less confident in giving a higher score.

**Justification For Why Not Lower Score:**

Only one reviewer engaged in discussion and they raised their score, which concurs with my decision too.

---

### Decision · Program_Chairs · 2024-01-16

Accept (poster)